# Reticulocalbin 2 as a Potential Biomarker and Therapeutic Target for Atherosclerosis

**DOI:** 10.3390/cells11071107

**Published:** 2022-03-25

**Authors:** Jing Li, Angela M. Taylor, Ani Manichaikul, John F. Angle, Weibin Shi

**Affiliations:** 1Department of Radiology and Medical Imaging, University of Virginia, Charlottesville, VA 22908, USA; jjeileen2006@hotmail.com (J.L.); jfa3h@virginia.edu (J.F.A.); 2Department of Medicine, University of Virginia, Charlottesville, VA 22908, USA; amt6b@hscmail.mcc.virginia.edu; 3Center for Public Health & Genomics, University of Virginia, Charlottesville, VA 22908, USA; am3xa@virginia.edu

**Keywords:** reticulocalbin 2, biomarker, atherosclerosis, high-density lipoprotein

## Abstract

Vascular inflammation initiated by oxidized lipoproteins drives initiation, progression, and even rupture of atherosclerotic plaques. Yet, to date, no biomarker is directly linked to oxidized lipid-induced vascular inflammation. Reticulocalbin 2 (RCN2) is a key regulator of basal and oxidized lipid-induced cytokine production in arterial wall cells. We evaluated the potential of circulating RCN2 to identify subjects with or at risk of developing atherosclerosis. Immunohistochemical analysis revealed abundant RCN2 expression in the endothelium and adventitia of normal arteries and in atherosclerotic lesions of both humans and mice. Atherosclerosis-susceptible C57BL/6 (B6) mice had higher plasma Rcn2 levels than resistant C3H mice. High-fat diet feeding raised plasma Rcn2 levels of both strains. In humans, patients with coronary artery disease (CAD) or peripheral artery disease (PAD) showed elevated serum RCN2 levels compared to healthy controls. In a cohort of 92 CAD patients, serum RCN2 exhibited a significant inverse correlation with HDL cholesterol and K+ levels and a trend toward association with white blood cell account, Na+, statin treatment, and diastolic blood pressure. HDL treatment suppressed Rcn2 expression in endothelial cells. This study suggests that circulating RCN2 is a potential non-invasive biomarker for identifying individuals with atherosclerosis and HDL protects against atherosclerosis by downregulation of RCN2 expression in endothelial cells.

## 1. Introduction

Atherosclerosis is the underlying pathologic cause of coronary artery disease (CAD), ischemic stroke, peripheral artery disease (PAD), and a fraction of chronic kidney disease [1]. It silently builds up in the wall of large and medial arteries throughout adult life. Most individuals with the disease have no overt clinical symptoms, and over 50% of those who die of heart attack are not even aware of the condition [1]. By the time when symptoms occur, atherosclerosis often has entered the advanced stage and become life threatening. Thus, the current prevention strategy for reducing cardiovascular risk has shifted to the accurate identification of individuals at risk of developing a first cardiovascular event and the early detection of subclinical atherosclerosis [2,3,4]. Early interventions with lifestyle changes and preventive medications reduce acute cardiovascular events, resulting in considerable improvement in quality of life as well as economic savings [5].

Vascular inflammation initiated by oxidation of lipoproteins trapped in the subendothelial space plays an important role in all stages of atherosclerosis, from fatty streak, fibrous plaque to complicated lesions [6]. Atherosclerosis begins with the accumulation of plasma lipoproteins, predominantly LDL, in the subendothelial space of the arterial wall [7]. The trapped LDL undergoes oxidative modification by vascular cells to become oxidized LDL. The latter stimulates vascular cells to produce inflammatory molecules, such as MCP-1, ICAM-1, E-selectin, and VCAM-1, which recruit monocytes and T lymphocytes into the subendothelium and transform monocytes to macrophages [8,9]. The latter takes up oxidized LDL to become foam cells, the hallmark of atherosclerosis. As macrophages accumulate and form foam cells, atherosclerotic plaques grow. Although atherosclerotic plaques may block blood flow, approximately two-thirds of major adverse cardiovascular events, such as heart attack and stroke, are caused by plaque rupture and the ensuing thrombosis [10]. Inflammation drives the transition from stable plaques to vulnerable ones. At sites of rupture, there are plenty of macrophages but a paucity of smooth muscle cells [10]. The production of cytokines by local vascular cells elicits a prothrombotic state and enhances thrombosis [6]. Thus, establishing biomarkers that are directly connected to oxidized lipid-induced vascular inflammation would represent a major step forward in the capacity to ascertain individuals at high risk of developing cardiovascular events.

Reticulocalbin 2 (RCN2) is a calcium-binding protein with a signal sequence, six EF-hands, and an ER-retention sequence [11]. It is a candidate gene for an atherosclerosis susceptibility locus on mouse chromosome 9 [12]. RCN2 is abundantly expressed in atherosclerotic lesions and the endothelium [12,13]. Congenic mice that express a reduced amount of Rcn2 mRNA in the aorta develop smaller atherosclerotic lesions. siRNA knockdown of Rcn2 results in dramatic reductions of both basal and oxidized lipid-induction cytokine expression in endothelial cells [12]. Rcn2 also shows a dramatic suppression of endothelial nitric oxide (NO) production in mice [13]. However, to our knowledge, there has been no prior systematic investigation of RCN2 protein levels in relation to CAD risk in humans. Thus, in the present study, we examined whether circulating RCN2 in the blood could differentiate human subjects with atherosclerosis or at high risk of developing atherosclerosis from those without atherosclerotic risk.

## 2. Materials and Methods

### 2.1. Ethical Approval

This study used existing serum samples and clinical lab data collected under the protocols approved by the Institutional Review Board for Health Sciences Research (IRB-HSR #: 14620 and 10181) of the University of Virginia Medical School. All work involving the use of animals was performed in accordance with the current National Institutes of Health guidelines and approved by the Institutional Animal Care and Use Committee (protocol #: 3109). C57BL/6J (B6) *Apoe*^−/−^ mice were generated from breeding pairs purchased from the Jackson Laboratory, and C3H/HeJ (C3H) *Apoe*^−/−^ mice were generated in our laboratory [14]. Mice were weaned at 3 weeks of age onto a chow diet, switched onto a Western diet containing 21% fat, 48.5% carbohydrate, 17% protein, and 0.2% cholesterol (TD 88137, Envigo) at 6 weeks of age, and remained on the diet for 2 weeks.

### 2.2. Mouse Plasma Samples

Blood was drawn from the retroorbital veins of female B6 and C3H *Apoe*^−/−^ mice under isoflurane anesthesia after an overnight fast. Ethylenediamine tetra acetic acid (EDTA) was used as an anticoagulant to prevent blood clotting. Blood samples were spun in 1.5-mL tubes at 12,000 rpm for 5 min, and resultant plasma was collected and stored at −80 °C before use. Mice were bled once before and once after 2 weeks on a Western diet.

### 2.3. Human Serum Samples

All serum samples and clinical data were pre-existing and obtained in a de-identified fashion from patients with CAD or peripheral artery disease (PAD), and from healthy individuals. All blood samples were collected under fasting conditions, processed according to a standard protocol, and stored at −80 °C before use. Briefly, venous blood was collected into a test tube untreated with any anticoagulant, allowed to clot at room temperature, and then centrifuged at 2000× *g* for 10 min to remove the clot. The resultant serum was stored at −80 °C before assay. Serum samples from 20 healthy male subjects, aged 61 ± 5 years (50–68), were purchased from BioServe Biotechnologies, Beltsville, MD, USA. Healthy subjects had no known significant health problems based on detailed demographic information, including family history for three generations, gold standard clinical exam, medical history, and complete phenotypic data. Serum samples from 31 PAD patients, aged 61 ± 9 years (44–77), were obtained at the University of Virginia before they underwent angioplasty or angiogram procedures.

### 2.4. The Coronary Assessment in Virginia Cohort (CAVA)

The CAVA cohort consists of CAD patients, predominantly white, who underwent a medically necessary cardiac catheterization and intravascular ultrasound at the University of Virginia. Patients were excluded if they had a current acute coronary syndrome, cancer of any type, autoimmune disease of any type, or on immunosuppressive therapy, prior organ transplantation, anemia, pregnancy, HIV infection, or no vessel suitable for intravascular ultrasounds. Serum samples of serial 92 patients, aged 60 ± 10 years (33–80, mean ± SD), were analyzed for RCN2 concentrations.

Patients in the CAVA cohort have demographic data, information on statin use, measures of risk markers of atherosclerosis including HDL and LDL cholesterol, triglyceride, glucose, insulin, blood pressure, white blood cell account, and composition, hsCRP (high-sensitivity C-reactive protein), body mass index (BMI), and atheroma burden.

### 2.5. Quantification of Soluble RCN2 Protein

Pre-existing human serum (no anticoagulant involved) and mouse plasma samples were analyzed for RCN2 concentrations by using sensitive enzyme-linked immunosorbent assay (ELISA) kits (Cat. #: SEH244Hu and SEH244Mu, USCN Life Science Inc, Wuhan, China) according to manufacturer’s instructions. Briefly, 60 µL of diluted samples (4×), standard, and blank were added to appropriate wells and incubated for 2 h at 37 °C. Following a 1-h incubation with detection reagent A and then B, substrate solution was added to quantify RCN2 concentrations. To minimize inter-plate assay variations, the same set of standards and controls were run for all human serum samples.

### 2.6. Immunohistochemical Analysis

Expression of Rcn2 protein in the arterial wall was detected with a rabbit anti-mouse polyclonal Rcn2 antibody (Cat. #: 10193-2-AP, 1:200 dilution, ProteinTech Group, Rosemont, IL, USA), which also has reactivity to human and rat RCN2 protein, as reported [13]. Human normal artery specimens were collected at autopsy from accidental deaths. Atherosclerotic arterial specimens were harvested during surgery from a patient with a thoracic aortic aneurysm. The aortic roots of wild-type B6 and B6-*Apoe*^−/−^ mice were prepared as previously reported [13]. Human samples were fixed in 10% buffered formalin, processed by using the standard histological technique and embedded in paraffin. Deparaffinized tissue sections were incubated with the primary antibody overnight at 4 °C and then with biotinylated goat anti-rabbit IgG antibody (Vector Laboratories). Immunoreactivity was visualized as brown color with the ABC (avidin-biotin complex) kit from Vector Laboratories.

### 2.7. Endothelial Cell Culture and Treatment

Endothelial cells were isolated from the aorta of B6 mice using our previously reported method [8]. After an overnight incubation with Dulbecco’s Modified Eagle Medium (DMEM) containing 1% fetal bovine serum (FBS), confluent cells were treated with 45 mg/dl of human HDL prepared from healthy individuals (MyBioSource, San Diego, CA, USA; Cat. # MBS143110), 12 mM KCl (Sigma, St. Louis, MO, USA; Cat. # P3911), or medium only for 24 h. Total RNA was then extracted from the cells isolated from 5 mice with a Qiagen RNeasy kit and reverse transcribed with an Invitrogen cDNA synthesis kit. A BioRad real-time PCR kit with SYBR Green was used to quantify the expression level of Rcn2 relative to Gapdh in triplicate, as reported [12].

### 2.8. Statistical Analysis

Data were presented as means ± SD. Student’s *t* test or ANOVA was used to evaluate the statistical significance for differences between the means of two or more groups in RCN2 concentrations or other measures. Linear regression analysis was performed to determine associations of RCN2 with other variables. Linear regression models were adjusted for age, sex, and race. *p*-values of ≤0.05 were considered statistically significant.

## 3. Results

Expression of Rcn2 in arteries with and without atherosclerosis. Immunohistochemical analysis showed the expression of Rcn2 antigen in the walls of normal arteries in both mice and humans, particularly abundant in the endothelial layer and the adventitia (Figure 1). Strong Rcn2 expression was observed in atherosclerotic lesions of both species (Figure 2).

Plasma Rcn2 levels of two *Apoe*^−/−^ mouse strains with different atherosclerosis susceptibility. B6 and C3H *Apoe*^−/−^ mice differ markedly in their susceptibility to atherosclerosis, with the former strain developing much larger atherosclerotic lesions than the latter [15]. Plasma Rcn2 concentrations were measured in female B6 and C3H *Apoe*^–/–^ mice before and after two weeks on the Western diet. B6 mice had higher plasma Rcn2 levels than C3H mice on the Western diet (37.3 ± 1.9 vs. 19.9 ± 1.4 ng/mL; *p* = 0.0007) (Figure 3). On the chow diet, B6 mice also had a higher plasma Rcn2 level (26.8 ± 5.4 vs. 9.8 ± 4.1 ng/mL), although the difference did not reach statistical significance (*p* = 0.056). Compared to the chow diet, the Western diet led to an elevation in plasma Rcn2 levels of C3H mice (*p* = 0.043). The plasma Rcn2 level of B6 mice was also elevated on the Western diet, although the rise did not reach statistical significance.

Elevated serum RCN2 levels in patients with atherosclerosis. Serum RCN2 levels were significantly different among male patients with CAD or PAD and healthy individuals. Patients with CAD had a significantly elevated RCN2 levels in comparison to healthy subjects (55.3 ± 3.9 vs. 18.5 ± 2.0 ng/mL; *p* < 0.001) (Figure 4). Elevated RCN2 levels were also observed in patients with PAD (93.0 ± 10.5 vs. 18.5 ± 2.0 ng/mL; *p* < 0.001). Compared to CAD patients, PAD patients had significantly higher RCN2 levels (*p* < 0.001).

Correlations of serum RCN2 with cardiometabolic traits. We examined associations of serum RCN2 concentrations with clinical and lab measures available for the CAVA cohort. A significant inverse correlation with HDL cholesterol levels was observed among 92 patients with covariate adjustment for age, sex, and race (β = −0.0556568; r = −0.226; *p* = 0.016) (Table 1; Figure 5).

A significant inverse correlation of RCN2 concentrations with serum K+ level was observed with adjustment for age, sex, and race (β = −1.258; r = −0.242; *p* = 0.009) (Table 1, Figure 5). Patients with a higher serum K+ level had a lower RCN2 concentration.

A suggestive inverse correlation with total white blood cell (WBC) account (β = −0.146; r = −0.199; *p* = 0.062) and serum Na+ (β = −0.139; r = −0.186; *p* = 0.053) levels was observed (Figure 5). A similar magnitude of correlation with blood neutrophil count was also observed (Table 1). We observed a trend towards a positive correlation with diastolic blood pressure (r = 0.149; *p* = 0.155), blood glucose (r = 0.123; *p* = 0.245), and triglyceride (r = 0.123; *p* = 0.244) and a negative correlation with BMI (r = −0.119; *p* = 0.257).

There was no correlation between RCN2 and LDL cholesterol levels (r = −0.055; *p* = 0.60) among the 92 CHD patients (Figure 5). However, we observed a trend towards a negative correlation between subjects treated with statins and those without in serum RCN2 and LDL cholesterol levels. Patients treated with statins tended to have lower RCN2 levels than those without statin treatment (r = −0.151; *p* = 0.152), so for LDL cholesterol (r = −0.151; *p* = 0.151).

Effects of HDL and K+ on Rcn2 mRNA expression in vitro. As shown above, serum RCN2 concentrations were significantly correlated with HDL cholesterol and K+ levels among CAD patients in the CAVA cohort. The influence of HDL and K+ on Rcn2 expression was evaluated in endothelial cells isolated from three and five B6 mice, respectively. Real-time PCR analysis showed that endothelial cells treated with HDL expressed significantly less Rcn2 transcript than cells treated with medium (Figure 6). The ratio of Rcn2 to Gapdh in the threshold cycle or Ct value was 0.061 ± 0.007 for HDL treated cells, significantly lower than that of 0.125 ± 0.011 for cells treated with medium only (*p* = 0.013). We measured Rcn2 concentrations in the medium that incubated with endothelial cells in the presence or absence of HDL. Because of the low sensitivity of the ELISA kit, the effect of HDL on Rcn2 expression was not detectable at the protein level (Appendix A under “HDL KCl on Rcn2_ELISA” worksheet).

Endothelial cells treated with K+ also expressed less Rcn2 transcript than cells treated with medium only (Rcn2/Gapdh ratio in Ct value: 0.077 ± 0.014 vs. 0.095 ± 0.018), although the difference did not reach statistical significance (*p* = 0.65).

## 4. Discussion

Inflammation plays an important role in the pathogenesis of atherosclerosis and its complications [16]. To date, C-reactive protein (CRP) is the only inflammatory biomarker used clinically for cardiovascular risk assessment and stratification. Increased CRP levels generally predict an increased risk of major cardiovascular events [17,18]. However, CRP is produced in the liver and not specific to atherosclerosis; rather, a high CRP level indicates the occurrence of an acute phase response, which can be induced by various inflammatory insults, such as infection, trauma, and allergy [19]. Thus, there is a clinical need for finding biomarkers capable of reflecting vascular inflammation in atherosclerosis. RCN2 is a key regulator of basal and oxidized lipid-induced inflammatory molecule production and also a potent inhibitor of endothelium-derived nitric oxide release [12,13]. Here we evaluated its potential as a novel biomarker for differentiating individuals with atherosclerosis from those without the condition. RCN2 was abundantly expressed in the endothelium and atherosclerotic plaques of both humans and mice. Patients with atherosclerosis and mice susceptible to atherosclerosis had elevated serum or plasma RCN2 levels. In a cohort consisting of CAD patients, serum RCN2 levels showed significant inverse associations with HDL cholesterol and K^+^ levels.

We found that serum RCN2 concentrations were significantly elevated in patients with atherosclerosis in comparison with healthy individuals. RCN2 expression was found abundantly expressed in atherosclerotic lesions (Figure 2). Thus, atherosclerotic plaques probably contributed to elevated serum RCN2 concentrations in patients with atherosclerosis. Patients with PAD were found to have higher serum RCN2 levels than CAD patients. An explanation for this elevation is that patients with PAD often have more extensive atherosclerotic plaques than CAD patients. Indeed, autopsy studies have shown that atherosclerotic plaques in peripheral arteries occur later than coronary plaques and when peripheral artery plaque has developed, coronary plaque often has progressed to a more advanced stage [20,21].

This study showed that atherosclerosis-susceptible B6-*Apoe*^−/−^ mice had higher plasma Rcn2 levels than resistant C3H-*Apoe*^−/−^ mice before and after two weeks on the Western diet. Because both *Apoe*^−/−^ strains should have no atherosclerotic lesions in the aorta and its branches at six weeks of age when the Western diet was initiated [22], other sources than atherosclerotic plaques should contribute to the difference of the two strains in plasma Rcn2 levels. One obvious source is the vascular wall where Rcn2 is abundantly expressed in the endothelium and adventitia. Indeed, we previously observed that B6 mice express more *Rcn2* transcript in the aorta than congenic mice carrying a C3H chromosome 9 region harboring the *Rcn2* gene [12]. After two weeks on the Western diet, B6-*Apoe*^−/−^ mice should have developed fatty streak lesions at the susceptible sites of the aorta but C3H-*Apoe*^−/−^ mice have not [14,22,23]. Abundantly expressed Rcn2 in atherosclerotic lesions probably contributed to elevated plasma Rcn2 levels in B6 mice on the Western diet. Although the plaque size of B6-*Apoe*^−/−^ mice at this time point was small, the contribution of plaque-derived Rcn2 to the rise in plasma Rcn2 levels could be sufficient in regard to the small volume of the blood (~1 mL).

Our analysis of the CAVA cohort consisting of CAD patients revealed a significant inverse association between serum RCN2 and HDL cholesterol levels. Association persisted after adjustments for age, sex, and race. Although HDL cholesterol has long been known for its protective effect against atherosclerosis and inflammation [24], the underlying mechanism remains unclear. In this study, we demonstrated that HDL suppressed endothelial expression of Rcn2, which is a key regulator of basal and oxidized lipid-induced inflammatory molecule production [12]. Thus, one likely mechanism by which HDL exerts anti-inflammatory and anti-atherosclerotic effects is through the suppression of Rcn2 expression in endothelial cells. Nevertheless, in vivo functional study is needed to determine whether Rcn2 mediates the anti-inflammatory and anti-atherosclerotic effects of HDL, including the use of gene knockout mice.

A significant inverse association was found between serum RCN2 and potassium levels in the CAVA cohort of CAD patients, even after adjustments for age, sex, and race. Emerging evidence has suggested that low serum K+ is associated with increased risk for metabolic disease and atherosclerosis [25,26,27]. Our finding indicated that the protective role of K+ is partially mediated by RCN2. Indeed, treatment with K^+^ suppressed Rcn2 expression in endothelial cells, although the effect was moderate. Given that atherosclerosis is a multifactorial chronic disease, the enhancement of low K^+^ to RCN2-mediated inflammation could be pathologically significant in vivo.

We observed a suggestive inverse correlation between serum RCN2 concentrations and blood WBC count in the CAVA cohort of CAD patients. An explanation for the finding is that most of the CAD patients had normal blood WBC counts (4000–11,000/μL). Even if a few had a high WBC count, it only suggests the occurrence of inflammation, which can be induced by hypercholesterolemia [28], infection, stress, obesity, and smoking [29]. The current commonly used biomarker CRP often shows no further rise in individuals with genetically elevated CRP levels by the increased risk of ischemic heart disease [30]. In contrast, blood RCN2 levels appeared to be more specific and related to atherosclerosis and risk factors.

Serum RCN2 levels showed a trend toward association with statin treatment in the CAVA cohort of CAD patients. Patients with statin treatment tended to have lower RCN2 levels than those without the treatment. As there was no correlation between RCN2 and LDL cholesterol levels, the association with RCN2 levels should be independent of the statins’ effect on LDL. Statins have been known to exert an atheroprotective role independent of the LDL-lowering effect [31]. The association observed here suggests a possibility that statins exert atheroprotective effects through action on RCN2.

We observed a trend towards association between serum RCN2 concentrations and blood pressure in the CAVA cohort. The human RCN2 gene is located in a chromosomal region where genetic variants are associated with blood pressure or hypertension (21). Deletion of the Rcn2 gene lowers blood pressure and attenuates angiotensin II-induced hypertension in mice [13].

One important alteration in endothelial dysfunction is an increased production and biological activity of the potent vasoconstrictor and pro-inflammatory peptide endothelin 1 [32]. However, its expression in the endothelial cells of mice was not affected as determined by Affymetrix microarray, although 49 inflammatory genes were down-regulated after Rcn2 was knocked down (unpublished data).

## 5. Conclusions

This study suggests the potential of circulating RCN2 as a novel biomarker for identifying individuals with atherosclerosis. The association with circulating RCN2 in a human cohort of CHD patients and the direct inhibition of Rcn2 expression in endothelial cells provide a mechanistic understanding of the anti-inflammatory and anti-atherosclerotic effects of HDL. This study has several limitations: firstly, male mice were not included. Sexual dimorphism in atherosclerosis is prominent, with females developing larger lesion sizes than their male counterparts [33]. Thus, findings achieved from female mice may be distinct from those observed from males. Secondly, the correlation between Rcn2 levels and lesion size was determined in mice. Thirdly, this study has not tested whether HDL isolated from CAD patients retains its capacity to inhibit RCN2 expression, although its inverse correlation with circulating RCN2 observed from the cohort of CAD patients suggests so.

## Figures and Tables

**Figure 1 cells-11-01107-f001:**
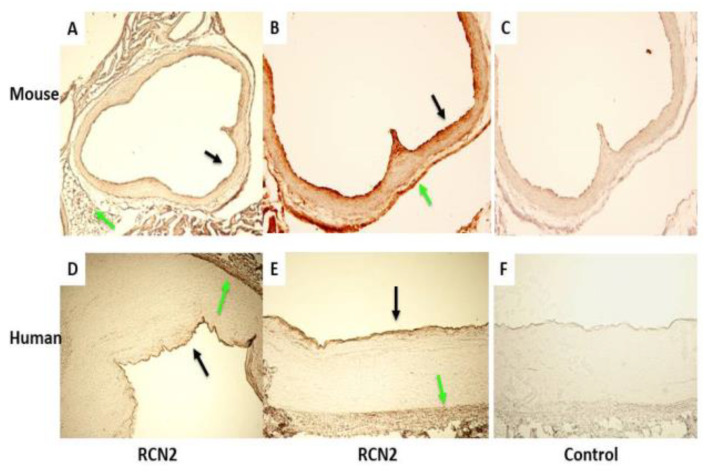
Immunohistochemical detection of reticulocalbin 2 (RCN2) expression in the normal arterial walls of a mouse and a human. Sections were stained with (**A**,**B**,**D**,**E**) or without (**C**,**F**) the presence of RCN2 antibody. Note strong RCN2 expression in the endothelial layer and adventitia of the artery wall. Original magnifications: ×4 (**A**,**D**), ×10 (**B**,**C**,**E**,**F**). Black arrows point at stained intima, and green arrows at stained adventitia.

**Figure 2 cells-11-01107-f002:**
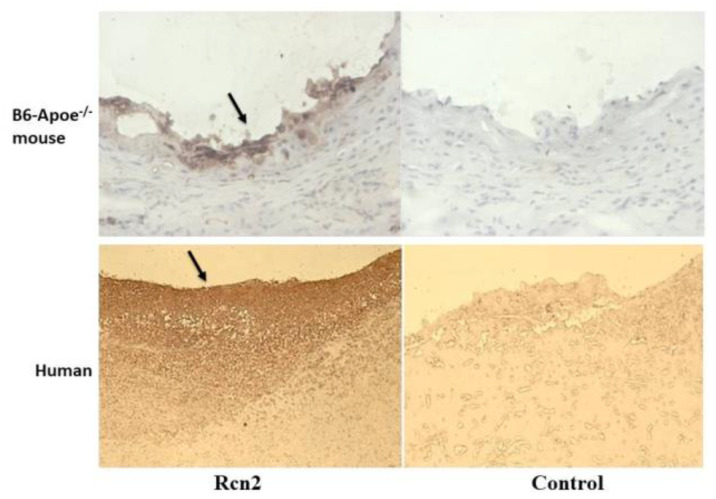
Immunohistochemical detection of RCN2 expression in atherosclerotic lesions of a mouse and a human. Sections were stained with (**left**) or without (**right**) the presence of RCN2 antibody. Note strong RCN2 expression in atherosclerotic lesions of both mouse and human. Black arrows point at stained atherosclerotic lesions. Original magnification: ×10 (**top** row) or ×4 (**bottom** row).

**Figure 3 cells-11-01107-f003:**
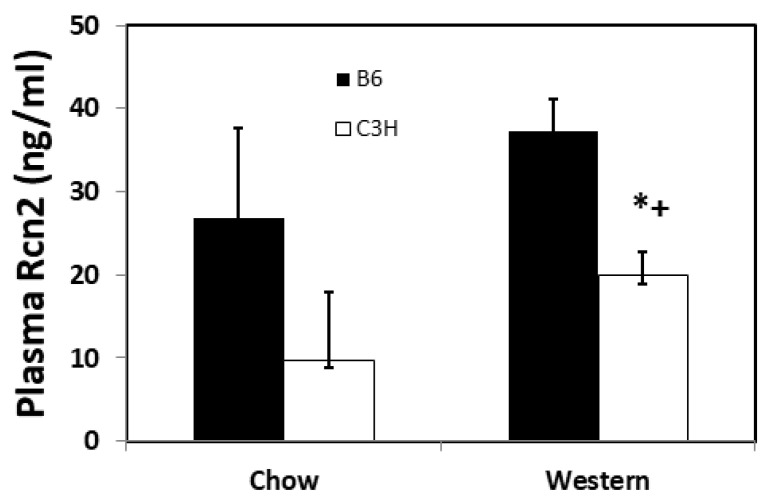
Plasma Rcn2 protein levels (ng/mL) of female B6 and C3H *Apoe*^−/−^ mice fed a chow or Western diet. Results are means ± SD of 4 mice per strain. * *p* < 0.05 versus B6 mice fed the same diet, and + *p* < 0.05 versus the same mouse strain fed the chow diet. Two-way ANOVA was performed to determine statistical significance in Rcn2 levels between the two strains on the two diets.

**Figure 4 cells-11-01107-f004:**
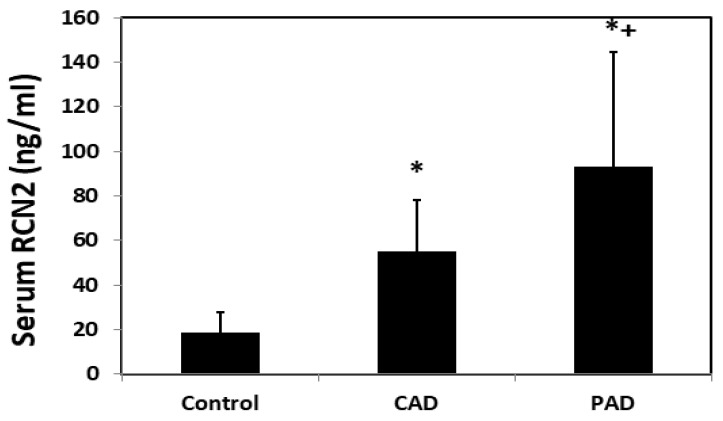
Serum RCN2 levels of patients with coronary artery disease (CAD) or peripheral artery disease (PAD) compared with healthy individuals. Results are means ± SD of 20 to 34 subjects per group. * *p* < 0.05 vs. healthy individuals. + *p* < 0.05 vs. CAD. One-way ANOVA was performed to determine statistical significance in RCN2 levels among the groups.

**Figure 5 cells-11-01107-f005:**
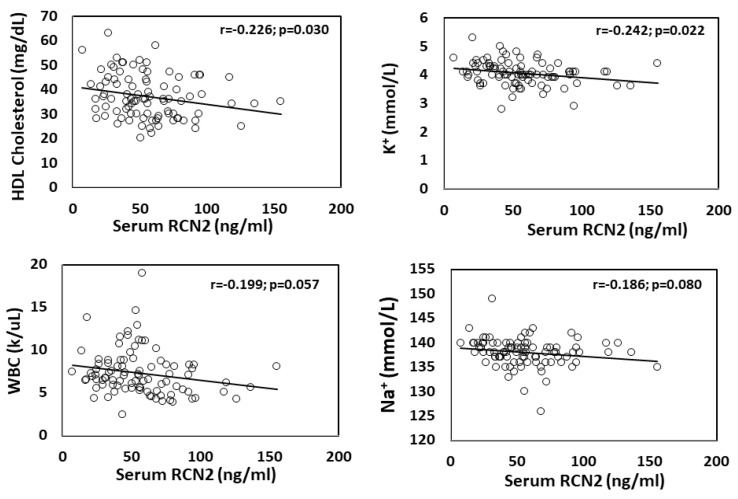
Associations of serum RCN2 concentrations with serum risk parameters relevant to cardiometabolic diseases in 92 patients with coronary heart disease. The regression line, P- and R-values were determined using the linear regression model. Serum RCN2 shows a significant inverse association with serum K+ and HDL cholesterol, a trend of association with WBC (white blood cells) count, Na+, triglyceride, diastolic BP (blood pressure), BMI, glucose and statin treatment, and no association with LDL. Association of hsCRP with atheroma burden, expressed as % of the maximal luminal diameter reduction, is also shown. Each dot represents an individual patient.

**Figure 6 cells-11-01107-f006:**
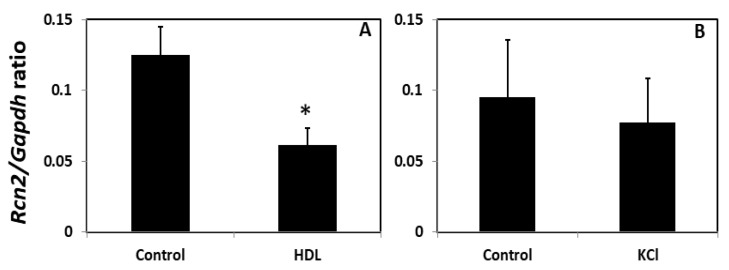
Real-time PCR analysis of Rcn2 expression (normalized to Gapdh) in endothelial cells isolated from B6-*Apoe*^−/−^ mice. Cells received a 24-h treatment with 45 mg/dl of HDL (**A**) or 12 mM of KCl (**B**) in final concentrations. Values are means ± SD of in cells from 3 to 5 mice. * *p* < 0.05. The Student *t* test was performed to determine statistical significance in Rcn2 expression levels in cells treated with and without HDL or KCl.

**Table 1 cells-11-01107-t001:** Regression analysis of associations of serum RCN2 concentrations with clinical and lab measures in the CAVA cohort.

Lab or Clinical Measure	N	Beta Estimate ^a^	SE ^b^	*p* Value
BP Systolic mmHG	92	0.00969245	0.01030022	0.349313076
BP Diastolic mmHG	92	0.01765266	0.01419677	0.217050465
Height (inches)	92	0.11660038	0.07012828	0.099976831
Weight (lbs)	92	−0.0011682	0.00440845	0.791642773
BMI	92	−0.0216007	0.02779423	0.439169159
Diabetes	92	0.03297067	0.41993906	0.937600302
Hypertension	92	−0.542929	0.50886634	0.288951539
Statin	92	−0.7332029	0.49621175	0.143125728
Na^+^ (mmol/L)	90	−0.1386291	0.07066807	0.053070947
K^+^ (mmol/L)	89	−1.2575188	0.4733591	**0.00944517**
BUN (mg/dL)	90	−0.0075988	0.03604409	0.833533559
Creatinine (mg/dL)	90	0.06394834	1.04243609	0.951228397
Glucose (mg/dL)	91	0.0044541	0.00404779	0.274236856
A1c (%)	66	0.08715481	0.16033648	0.588715504
eGFR using CKD-EPI mL/min per 1.73 m^2^	90	−0.0029378	0.01386449	0.832699896
eGFR using MDRD mL/min per 1.73 m^2^	90	−0.0005513	0.01323238	0.96686373
WBC (k/uL)	92	−0.1457938	0.07714427	0.062103295
Hgb (k/uL)	92	0.02462452	0.0927292	0.791211679
HCT (%)	92	0.0094704	0.042752	0.825207044
Platelets (k/uL)	92	−0.0042103	0.00369879	0.258127884
Neutrophils Percent	74	−0.0140214	0.01932898	0.470655622
Neutrophils Absolute Count (k/uL)	74	−0.1894186	0.09953278	0.061203044
Lymphocytes Percent	74	0.0288674	0.02335576	0.220654752
Lymphocytes Absolute Count (k/uL)	74	−0.353782	0.29906492	0.240884786
Monocytes Percent	74	0.02998299	0.06812356	0.661221099
Monocytes Absolute Count (k/uL)	74	−1.2669083	0.86871079	0.149273558
Eosinophils Percent	74	−0.0171623	0.12760792	0.89340498
Eosinophils Absolute Count (k/uL)	74	−1.9731085	1.63316999	0.231115393
Basophils Percent	74	0.09109683	0.44496708	0.838388038
Basophils Absolute Count (k/uL)	74	−4.1208846	3.82795515	0.285442693
Total Cholesterol (mg/dL)	88	−0.0021	0.00578931	0.717727569
Triglycerides (mg/dL)	92	0.0035148	0.00232033	0.133450441
HDL Cholesterol (mg/dL)	92	−0.0556568	0.02265705	**0.016016942**
LDL Cholesterol (mg/dL)	92	−0.0008449	0.00629427	0.893532828
hsCRP (mg/L)	42	−0.0174402	0.03398638	0.610895114
Insulin (ulU/mL)	42	−0.0080212	0.02384196	0.738445545

^a^ Estimation of the strength of an association of serum RCN2 with a lab or clinical measure after adjustment for age, sex, and race. ^b^ Standard error. Significant association is denoted in bold. Raw data are available in Appendix A under “IVUS_correlation”. BP: blood pressure; BMI: body mass index; Na: sodium ion; K: potassium ion; BUN: blood urea nitrogen; A1c: glycated hemoglobin; eGFR: estimated glomerular filtrate rate; CKD-EPI: chronic kidney disease epidemiology collaboration; MDRD: Modification of Diet in Renal Disease; WBC: white blood cells; Hgb: hemoglobin; HCT: hematocrit; hsCRP: high-sensitivity C-reactive protein.

## Data Availability

All data included in this article can be found in Appendix A.

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
