# Peer review of "Reticulocalbin 2 as a Potential Biomarker and Therapeutic Target for Atherosclerosis"

_cells, 2022, doi:10.3390/cells11071107_

Round 1

Reviewer 1 Report

In this manuscript, Li et al determined reticulocalbin 2 (RCN2) expression levels in vascular walls in human and mice. Their results reveal that RCN2 are abundantly expressed in the endothelium and adventitia in either normal arteries or in atherosclerotic lesions. Interestingly the results show that atherosclerosis-susceptible C57BL/6 mice have higher plasma RCN2 levels than those in resistant C3H mice and that high fat diet feeding raised plasma RCN2 levels in both mouse strains. The authors also evaluated RCN2 serum levels in patients with coronary artery disease (CAD) or peripheral artery disease (PAD) compared to healthy controls and found that serum RCN2 levels were elevated in CAD and PAD cases compared to healthy controls. Another interesting finding is that serum RCN2 levels are in inverse association with HDL levels, potassium levels and statin treatment. The in vitro experiments demonstrated that HDL treatment lowed RCN2 expression in ECs. These results suggest that RCN2 is a potential new biomarker/therapeutic target for atherosclerosis. These findings are new and interesting in the field of atherosclerosis.

There are some concerns:

  1. Why were plasma RCN2 levels measured in mice, but serum RCN2 levels were measured in human samples? The method for serum collection needs to be documented, not only “standard protocol” mentioned in line 93.
  2. The method for RCN2 measurement needs to be described in Fig. 3.
  3. One sub-figure of Fig. 5 is left without labeling of the x- axis.
  4. RCN2 and Rcn2 were randomly presented.
  5. Na+ and K+ should be correctly presented in Table 1.

Author Response

Response to Reviewer 1

Comment: Why were plasma RCN2 levels measured in mice, but serum RCN2 levels were measured in human samples? The method for serum collection needs to be documented, not only “standard protocol” mentioned in line 93.

Response: Amended. Human serum samples were prepared without using any anticoagulant, while mouse plasma samples were prepared using EDTA as an anticoagulant.

Comment: The method for RCN2 measurement needs to be described in Fig. 3.

Response: Amended.

Comment: One sub-figure of Fig. 5 is left without labeling of the x- axis.

Response: Amend.

Comment: RCN2 and Rcn2 were randomly presented.

Response: According to the nomenclature, human protein symbols are capitalized, with all letters in uppercase, and mouse protein symbols generally are only the first letter in uppercase and the remaining letters in lowercase. Gene symbols are italicized for both human and mouse.  

Comment: Na+ and K+ should be correctly presented in Table 1.

Response: Corrected.

Reviewer 2 Report

In this manuscript the authors provide preliminary evidence that supports a potential role of  reticulocalbin 2 (RCN2), as a clinically useful biomarker of atherosclerosis. First they showed that RCN2 is expressed abundantly in the arterial wall of mice and humans and that its expression in increased in  atherosclerotic plaques. Then they measured RCN2 levels in a cohort of CAD patients and they found correlations of serum RCN2 with HDL cholesterol and K+ levels. Finally they show that treatment of endothelial cells with HDL or KCl caused a reduction in RCN2 mRNA levels in line with their observations in humans. They conclude that RCN2 could represent a novel biomarker for identifying individuals with atherosclerosis and could be clinically exploited.

Overall, the findings are novel and the conclusions are supported by the data. There are some issues that the authors should address before publication is recommended:

1.The circulating levels of RCN2 in mice, shown in Fig. 3, was performed in female mice only that received a chow diet or a western diet for 2 weeks. Why did the authors use only female mice while it is known that male mice are more prone to diet-induced atherosclerosis and why the mice were fed with HFD for 2 weeks only and not for longer periods for the development of more advanced lesions? Is there any correlation between RCN2 levels and lesion size in mice?

2.The authors state (lines 167-168) that B6 mice had higher plasma Rcn2 levels than C3H mice on both chow and WD. However,  in Fig. 3 the difference in RCN2 levels between these two mouse strains under chow diet is not statistically significant. The authors should rewrite this statement.

3.In Fig. 6 the authors show that treatment of mouse endothelial cells with HDL caused a reduction in RCN2 mRNA levels. Based on this finding, they suggest that one likely mechanism by which HDL exerts its anti-inflammatory and anti-atherosclerotic effects is through the suppression of RCN2 in endothelial cells (lines 293-294). However, it is known that patients with CAD  are characterized by dysfunctional HDL. Have the authors tested whether HDL isolated from CAD patients retains its capacity to inhibit RCN2 expression?

4. In Fig. 5 it is shown that there is a negative correlation between RCN2 levels and statin use suggesting that statins protect from CAD by decreasing RCN2 expression presumably in endothelial cells where it is physiologically expressed. The authors could support this statement by a simple experiment i.e. treatment of endothelial cells with statin or vehicle and comparing the levels of endogenous RCN2.

Author Response

Response to Reviewer 2.

Comment 1. “The circulating levels of RCN2 in mice, shown in Fig. 3, was performed in female mice only that received a chow diet or a western diet for 2 weeks. Why did the authors use only female mice while it is known that male mice are more prone to diet-induced atherosclerosis and why the mice were fed with HFD for 2 weeks only and not for longer periods for the development of more advanced lesions? Is there any correlation between RCN2 levels and lesion size in mice?

Response: Female mice were used because they are more susceptibility to atherosclerosis than male counterparts in dietary, Apoe-/- and Ldlr-/- models (Circulation Research 2020;126:1297. Atherosclerosis 1987;64:215. Arteriosclerosis and Thrombosis 1994;14:1480). After 2 weeks on the Western diet, B6-Apoe-/- mice have developed fatty streak lesions at the susceptible sites of the aorta but C3H-Apoe-/- mice have not (atherosclerosis 1994; 14: 133. J Vasc Res 2019; 56: 241). The data we have does not allow us to define the correlation between RCN2 levels and lesion size in mice.  We have also addressed the limitations in Discussion.   

Comment 2.The authors state (lines 167-168) that B6 mice had higher plasma Rcn2 levels than C3H mice on both chow and WD. However, in Fig. 3 the difference in RCN2 levels between these two mouse strains under chow diet is not statistically significant. The authors should rewrite this statement.

Response: Revised.

Comment 3. In Fig. 6 the authors show that treatment of mouse endothelial cells with HDL caused a reduction in RCN2 mRNA levels. Based on this finding, they suggest that one likely mechanism by which HDL exerts its anti-inflammatory and anti-atherosclerotic effects is through the suppression of RCN2 in endothelial cells (lines 293-294). However, it is known that patients with CAD are characterized by dysfunctional HDL. Have the authors tested whether HDL isolated from CAD patients retains its capacity to inhibit RCN2 expression?

Response: The HDL used to treat endothelial cells was purchased from MyBioSource, which was prepared from healthy individuals.  We have not tested whether HDL isolated from CAD patients can inhibit RCN2 expression.  The inverse relationship between HDL and RCN2 was observed in a cohort of CAD patients suggests that HDL retains its athero-protective role.  We have addressed this issue in Discussion.

Comment 4. In Fig. 5 it is shown that there is a negative correlation between RCN2 levels and statin use suggesting that statins protect from CAD by decreasing RCN2 expression presumably in endothelial cells where it is physiologically expressed. The authors could support this statement by a simple experiment i.e. treatment of endothelial cells with statin or vehicle and comparing the levels of endogenous RCN2.

Response: We thank the reviewer for the valuable comment and will perform the suggested study in the future.  Such study can’t be accomplished within 10 days, the time allowed to revise the manuscript.

Reviewer 3 Report

Reviewer comments:

Authors work aimed to evaluate the role of reticulocalbin 2 (RCN2) as a potential circulating biomarker for identifying individuals with atherosclerosis or at high risk to develop the disease. Main results are: RCN2 was found to be highly expressed in the endothelium and adventitia of normal artery and atherosclerotic lesions, HFD increased plasma levels of RCN2 in mice, high levels of RCN2 were also observed in patients with coronary artery disease (CAD) or peripheral artery disease (PAD), and a significant inverse correlation of serum RCN2 with HDL cholesterol and K+ levels was observed in CAD patients. Further analysis revealed that HDL treatment of endothelial cells from mice suppressed RCN2 expression. Based on all these findings, authors concluded that circulating RCN2 could be a potential non-invasive biomarker for identifying individuals with atherosclerosis.

Overall, the preliminary data presented in this manuscript are highly interesting but the experimental design definitely needs some improvements to complete the study.

Major comments:

  1. Reviewer worries about the weak statistical power in this study. This is probably due to the small sample size, for instance, in Fig 3, only 4 mice were used per experiment. Therefore, authors are encouraged to improve the statistical significance by increasing the size of studied animals or subjects.
  2. Authors concluded that RCN2 could to be a potential biomarker for atherosclerosis but this assumption has to be confirmed and validated by further appropriate experimental design. In this context, targeting RCN2 gene (RCN2 knock out, siRNA, …) in animal/cells model is recommended. This approach may bring additional valuable information to the study.
  3. Authors postulate that HDL suppresses endothelial expression of RCN2 to protect against atherosclerosis and inflammation. I am not sure whether we have here enough information to make such a conclusion. Authors are encouraged to comment on possible mechanisms by which HDL may suppress RCN2 and how these can be examined in future studies. Reviewer comment in 2. (RCN2 knockdown) may be one of the strategies that can help clarify these questions.

Minor comments:

- Page 1, line 24, “ … non-invative…”, please change to noninvasive

- “Figure 6. Real-time PCR analysis of Rcn2 expression relative to Gapdh in endothelial cells isolated 239 from B6-Apoe-/- mice after a 24-h treatment with 45 mg/dl of HDL (A) or 12 mM of KCl (B) in final 240 concentrations.” Please rephrase the title of this figure because it gives the impression that B6-Apoe-/- mice are treated with HDL and not cells in culture.

Author Response

Reviewer 3

Major comments:

Comment 1: Reviewer worries about the weak statistical power in this study. This is probably due to the small sample size, for instance, in Fig 3, only 4 mice were used per experiment. Therefore, authors are encouraged to improve the statistical significance by increasing the size of studied animals or subjects.

Response: We agree with the reviewer’s comment. We were requested to revise the manuscript within 10 days and thus have no time to perform additional experiments.  As the reviewer may have noticed, the difference between the two strains in plasma Rcn2 levels was observed on both chow and Western diets so the significance at the biological level should be out of question.      

Comment 2: Authors concluded that RCN2 could to be a potential biomarker for atherosclerosis but this assumption has to be confirmed and validated by further appropriate experimental design. In this context, targeting RCN2 gene (RCN2 knock out, siRNA, …) in animal/cells model is recommended. This approach may bring additional valuable information to the study.

Response: We thank the reviewer for the valuable comment. In published study, we knocked down Rcn2 in endothelial cells and demonstrated its key role in regulating both basal and oxidized lipid-induced cytokine production (Am J Physiol Heart Circ Physiol. 2011 Sep;301(3):H1056 ).  We did test the effect of HDL and KCl on Rcn2 expression in endothelial cells at the protein level.  Because of the low sensitivity of the ELISA kit, the effect was not detectable at the protein level.  These results are presented in Supplemental data under “HDL KCl on Rcn2_ELISA” worksheet and also amended in Results.  

Comment 3: Authors postulate that HDL suppresses endothelial expression of RCN2 to protect against atherosclerosis and inflammation. I am not sure whether we have here enough information to make such a conclusion. Authors are encouraged to comment on possible mechanisms by which HDL may suppress RCN2 and how these can be examined in future studies. Reviewer comment in 2. (RCN2 knockdown) may be one of the strategies that can help clarify these questions.

Response: As shown in Figure 6, treatment with HDL suppressed Rcn2 expression in endothelial cells.  We previously reported that Rcn2 is a key regulator of both basal and oxidized lipid-induced cytokine production in endothelial cells (Am J Physiol Heart Circ Physiol. 2011 Sep;301(3):H1056 ).

Minor comments:

Comment- Page 1, line 24, “ … non-invative…”, please change to noninvasive

Response: Corrected.

Comment: “Figure 6. Real-time PCR analysis of Rcn2 expression relative to Gapdh in endothelial cells isolated 239 from B6-Apoe-/- mice after a 24-h treatment with 45 mg/dl of HDL (A) or 12 mM of KCl (B) in final 240 concentrations.” Please rephrase the title of this figure because it gives the impression that B6-Apoe-/- mice are treated with HDL and not cells in culture.

Response: revised.

Round 2

Reviewer 2 Report

The authors provided satisfactory answers to all my comments.

Author Response

We thank the reviewer for your time and effort in reviewing our manuscript. 

Reviewer 3 Report

Dear Editor,

The authors of this manuscript have addressed some on my questions related to their studies, but not comment # 3.

The comment was as follow:

“Authors are encouraged to comment on possible mechanisms by which HDL may suppress RCN2 and how these can be examined in future studies. “

The authors response was: 

"As shown in Figure 6, treatment with HDL suppressed Rcn2 expression in endothelial cells.  We previously reported that Rcn2 is a key regulator of both basal and oxidized lipid-induced cytokine production in endothelial cells (Am J Physiol Heart Circ Physiol. 2011 Sep;301(3):H1056 )."

The above reference was checked carefully, however, I could not find the answer to the question. I recommend that the authors complete the manuscript by providing the answer to such a question that is obvious.

Author Response

Comment: “Authors are encouraged to comment on possible mechanisms by which HDL may suppress RCN2 and how these can be examined in future studies. “The authors response was: "As shown in Figure 6, treatment with HDL suppressed Rcn2 expression in endothelial cells.  We previously reported that Rcn2 is a key regulator of both basal and oxidized lipid-induced cytokine production in endothelial cells (Am J Physiol Heart Circ Physiol. 2011 Sep;301(3):H1056 )." The above reference was checked carefully, however, I could not find the answer to the question. I recommend that the authors complete the manuscript by providing the answer to such a question that is obvious.

Response: In the published study in Am J Physiol Heart Circ Physiol (2011;301:H1056), we reported the critical role of Rcn2 in both basal and oxidized lipid-induced proinflammatory gene expression in endothelial cells, which was discovered through siRNA knockdown.  Like siRNA, HDL suppressed Rcn2 expression in endothelial cells so expected are the downstream proinflammatory genes.  We agree with the reviewer, in vivo functional study is needed to determine whether Rcn2 mediates the anti-inflammatory and anti-atherosclerotic effects of HDL, including the use of gene knockout mice.  We have amended the reviewer’s comment in Discussion.